# The Effect of FG-Nup Phosphorylation on NPC Selectivity: A One-Bead-Per-Amino-Acid Molecular Dynamics Study

**DOI:** 10.3390/ijms20030596

**Published:** 2019-01-30

**Authors:** Ankur Mishra, Wouter Sipma, Liesbeth M. Veenhoff, Erik Van der Giessen, Patrick R. Onck

**Affiliations:** 1Zernike Institute for Advanced Materials, University of Groningen, Groningen, 9747 AG, The Netherlands; a.mishra@rug.nl (A.M.); woutersipma@gmail.com (W.S.); e.van.der.giessen@rug.nl (E.V.G.); 2European Research Institute for the Biology of Ageing, University of Groningen, University Medical Centre, Groningen, 9713 AV, The Netherlands; l.m.veenhoff@rug.nl

**Keywords:** Nuclear pore complex, FG-Nups, phosphorylation

## Abstract

Nuclear pore complexes (NPCs) are large protein complexes embedded in the nuclear envelope separating the cytoplasm from the nucleoplasm in eukaryotic cells. They function as selective gates for the transport of molecules in and out of the nucleus. The inner wall of the NPC is coated with intrinsically disordered proteins rich in phenylalanine-glycine repeats (FG-repeats), which are responsible for the intriguing selectivity of NPCs. The phosphorylation state of the FG-Nups is controlled by kinases and phosphatases. In the current study, we extended our one-bead-per-amino-acid (1BPA) model for intrinsically disordered proteins to account for phosphorylation. With this, we performed molecular dynamics simulations to probe the effect of phosphorylation on the Stokes radius of isolated FG-Nups, and on the structure and transport properties of the NPC. Our results indicate that phosphorylation causes a reduced attraction between the residues, leading to an extension of the FG-Nups and the formation of a significantly less dense FG-network inside the NPC. Furthermore, our simulations show that upon phosphorylation, the transport rate of inert molecules increases, while that of nuclear transport receptors decreases, which can be rationalized in terms of modified hydrophobic, electrostatic, and steric interactions. Altogether, our models provide a molecular framework to explain how extensive phosphorylation of FG-Nups decreases the selectivity of the NPC.

## 1. Introduction

Eukaryotic cells are characterized by the presence of the nuclear envelope (NE), a lipid bilayer membrane that separates the cells into two compartments, i.e., the nucleus and the cytoplasm. The NE contains many nuclear pore complexes (NPCs), which are the sole gateway for the exchange of essential biomolecules between the two compartments. NPCs are large protein complexes, with a molecular mass of ~ 55–66 MDa [1,2] in yeast and ~ 125 MDa in vertebrates [3]. The NPC is composed of 30 different types of proteins called nucleoporins (Nups) [4,5]. One third of these Nups are intrinsically disordered proteins (IDPs), which are anchored to the inner wall of the NPC and are rich in phenylalanine-glycine (FG) repeats. Inside the NPC, these FG-Nups form a central meshwork that provides a permeability barrier for translocating molecules. Various studies have revealed that the NPC allows rapid transport of small molecules (30 kDa or ~5 nm in diameter), but drastically slows down the translocation of larger molecules from one compartment to the other [6,7,8]. It also has been found that FG Nups bind to nuclear transport receptors (NTRs) [9,10] by means of hydrophobic interactions, which facilitates the translocation of NTRs by lowering the permeability barrier [11]. Cargoes of diameter up to 40 nm are known to translocate by this facilitated transport mechanism [12,13]. Therefore, the FG-Nups are considered to be crucial in establishing the selective permeability barrier of the NPC. 

Nucleocytoplasmic trafficking can be altered by a change in the surface properties of translocating molecules [14], by the deletion of FG-Nups [6,8], and by the change in cohesiveness of the FG-Nups [15]. For example, mutation of the hydrophobic F residues of Nsp1, a representative yeast FG-Nup, into the hydrophilic Serine S reduces the propensity of Nsp1 to form a hydrogel [16]. These experiments revealed that the hydrogels exclude inert molecules, but allow hydrophobic NTRs to enter, which is explained in terms of a local disruption of the cohesive gel network [16,17]. In a separate study [15], the Nsp1 molecules tethered onto the inner surface of solid state NPC mimics formed a dense phase (over 100 mg/mL) and enabled transport selectivity. Kap95 (a yeast NTR) traversed the pore whereas the translocation of tCherry (an inert molecule of similar size) was inhibited. The F, I, L, V to S mutation of Nsp1 resulted in a remarkably less dense FG-Nup network inside the pore, which led to a loss of selectivity, as both tCherry and Kap95 were able to translocate. Taken together, these studies show that the transition from a dense, hydrophobic phase to a dispersed, hydrophilic phase results in the nanopores losing their selective barrier function. 

The hydrophobicity of FG-Nups can also be altered through phosphorylation, one of the most abundant protein modifications inside the cell [18,19]. Phosphorylation is catalyzed by kinases and can be reversed by phosphatases. It has been shown that extracellular signal-regulated kinase (ERK), a phosphorylating agent, can directly interact with FG-Nups [20,21], causing FG-Nups to phosphorylate [22,23,24]. Several in vitro studies have revealed that specifically Nup62, Nup98, Nup153, Nup214, and Nup358 can undergo phosphorylation [25,26]. Furthermore, there is evidence which confirms that FG-Nups undergo phosphorylation in vivo as well [21,23,27,28]. Transport studies demonstrated that the phosphorylation of nucleoporins results in decreased kinetics of active transport of Kap95 [25,27,29] and Kap-cargo complexes [30,31], and increased kinetics of passive transport [32]. These studies indicate that phosphorylation can modulate the selective permeability of the NPCs. However, the molecular mechanism behind the alteration in nucleocytoplasmic transport due to phosphorylation is not well understood. 

Molecular dynamics (MD) simulations have proved to be a powerful tool to study the disordered protein structure inside the NPC and the transport through native and biomimetic nanopores [8,15,33,34,35]. Therefore, in order to understand the molecular mechanism behind the phosphorylation-induced alteration in transport kinetics, we carried out MD studies using our earlier developed one-bead-per-amino-acid (1BPA) coarse grained (CG) model for FG-Nups [35], extended here for phosphorylated FG-Nups. This 1BPA model has been successfully applied to probe the (doughnut-like) density distribution of the disordered domain of yeast NPCs [35], the facilitated transport of NTRs through yeast [33] and biomimetic [15] NPCs, and the size selectivity for passive transport [6], in good agreement with experiments. Although the transport experiments on phosphorylated NPCs cited above were carried out on mammalian NPCs, we here used the yeast NPC model, which has structural and functional similarities to the vertebrate NPC [36]. 

In the current study, we extended our 1BPA model to phosphorylated FG-Nups and carried out MD simulations of FG-Nups in isolation, as well as within the NPC. We studied the impact of phosphorylation on the structure of the disordered phase and the transport across the NPC in two scenarios. In the first (referred to as the Phos_N scenario), we used the NetPhosYeast 1.0 server [37] to obtain the phosphorylated residues of the yeast FG-Nups (yielding phosphorylated serine (S) and threonine (T) residues only), and in the second (referred to as the Phos_Max scenario), we assumed that all phosphorylatable residues (serine (S), histidine (H), threonine (T), and tyrosine (Y)) were phosphorylated. We investigated the changes in conformation of phosphorylated FG-Nups compared to FG-Nups in their native state by using the Stokes radius ($R_{S}^{}$) as a measure for their size (see Section 2.1). We found that phosphorylation causes FG-Nups to extend by an amount that depends on the fraction of phosphorylatable residues and positively charged residues. In Section 2.2, we present a study on the collective interaction of phosphorylated FG-Nups inside the confined environment of the NPC. We found that phosphorylation drastically reduces the FG-Nup density inside the NPC. Finally, in Section 2.3 we report on simulation results of the phosphorylation-affected transport of inert particles and Kap95, and discuss these results in light of the various contributions to the interaction energy inside the NPC. Our transport simulations are in qualitative agreement with the experimentally-observed increase and decrease in transport rate of the passive and active transport pathways, respectively. Note that the Phos_N scenario predicts more phosphorylation sites than other phosphorylation databases, such as the fungi phosphorylation database (FPD), which provides a comprehensive list of experimentally validated phosphorylation sites [38]. The prediction from the FPD database is incorporated in the Appendix A (see the section “Sensitivity analysis”) to provide a scenario for experimentally validated phosphorylation sites. It is important to note that it is unclear which phosphosites predicted in either scenario are phosphorylated simultaneously in vivo, and hence the predictions provided in this study are not meant to mimic specific biological conditions, but rather to shed light on the fundamental mechanisms underlying the changes in transport kinetics of phosphorylated NPCs.

## 2. Results

In order to study the effect of phosphorylation on FG-Nups, we started with our previously developed MD model for intrinsically disordered proteins (IDPs) in their native state, coarse-grained at a resolution of one bead per amino acid (1BPA) [35]. This 1BPA model accounts for non-bonded hydrophobic and electrostatic interactions between the amino acids, including the effect of solvent polarity and ionic screening to mimic the solvent conditions inside the NPC. The model is accurate (within 20% error) in predicting the Stokes radius $R_{S}^{}$ [35] for a range of FG-Nups and FG-Nup segments [39]. In the current study, we extended the model for phosphorylation by accounting for the change in hydrophobicity and charge of four amino acids: S, H, T, and Y. We used a weighted average scheme of five predictor programs KOWWIN, ClogP, ChemAxon, ALOGPS, and miLogP [40,41,42,43] to predict the change in hydrophobicity due to the change in the chemical structure. For details on the model development for phosphorylated FG-Nups we refer to the Materials and Methods section (Section 4.2). The new parameters for phosphorylated amino acids are summarized in Table 1.

### 2.1. Effect of Phosphorylation on Isolated FG Nups

We used our newly developed parametrization for phosphorylation and performed MD simulations to study the effect of phosphorylation on the conformation of isolated FG-Nup segments [35]. The simulated trajectories were analyzed to determine the time averaged $R_{S}^{}$ using the Hydro program [44,45]. The predicted $R_{S}^{}$ values for the phosphorylated FG-Nups are compared with that of FG-Nups in their native state (from experiments [39] and simulations [35]) in Figure 1. The error bars for the simulation data represent the standard deviation in time for $R_{S}^{}$ (See Appendix A in the Appendix A for the source data).

As a result of phosphorylation, the amino acids become more hydrophilic and negatively charged (see Table 1). Thus, compared to the native state, the phosphorylated FG-Nups exhibit enhanced electrostatic repulsion and reduced hydrophobic attraction leading to an overall decrease in intra-molecular cohesion and thus a more extended configuration (see Figure 1). In Appendix A, we have summarized the number of amino acids that can be phosphorylated in each FG-Nup segment. The FG-segments are grouped as low charged (*lc*), high charged (*hc*), and stalk (*s*) domains, following the definition of Yamada et al. [39]. We found that the relative abundance of phosphorylatable residues in all FG-Nup segments ranges from ~15% (for Nup116s) to ~33% (for Nsp1n_lc) for the maximally phosphorylated (Phos_Max) condition, whereas for the Phos_N scenario the range is from ~4% (Nup116s) to ~17% (Nup159_hc). In order to quantify the change in Stokes radius in terms of the number of residues undergoing phosphorylation, we plot the normalized change in $R_{S}^{}$ as a function of the percentage of phosphorylatable residues (*n*) for the low charged, high charged, and stalk groups in blue, red, and green data points, respectively, for the Phos_Max (Figure 2a) and Phos_N (Figure 2b) scenarios. The change $\Delta R_{S}^{}$ is normalized as $\Delta R_{S}^{}/\left(N-1\right)b$, where *N* is the total number of residues of the FG-Nup segment and *b* is the coarse-grained bond length (3.8 Angstrom). We fitted the data points for individual groups to a straight line passing through the origin, represented as colored lines in Figure 2a,b. We note that for both Phos_Max and Phos_N, the FG-segments from the *lc* group show the highest normalized change in $R_{S}^{}$ (blue line in Figure 2a,b), whereas phosphorylation has a smaller effect on size for the *hc* and *s* groups (red and green lines in Figure 2a,b).

In order to investigate the varying response for the three groups, as shown in Figure 2a,b, we analyzed the change in hydrophobicity upon phosphorylation and found that it is roughly similar for the three groups, i.e., for FG-Nups from the *lc*, *hc*, and s groups, the hydrophobicity drops by 13-20%, 16–22%, and 12–21%, respectively, for Phos_Max (see Appendix A). Similarly, for Phos_N, the reduction in hydrophobicity amounts to 3–10%, 7–10%, and 4–12% for the *lc*, *hc*, and *s* groups, respectively, showing no major difference across the three groups. Clearly, the effect of phosphorylation on hydrophobicity alone cannot account for the different $R_{S}^{}$ of the groups. It has been argued that the net proline content in IDPs plays an important role in determining the effective Stokes radius [47], as proline provides additional stiffness to the peptide chain because of its ring structure. This effect of proline is included in our 1BPA model in the form of the bonded potentials [46]. However, all the FG-Nup segments analyzed in this study (Figure 1) have a similar 3–8% proline content, and therefore cannot explain the different Stokes radii across the three groups (see Appendix A). Next, we analyzed the effect of charge. Since the net charge of the three families is quite similar (i.e., 2–3%, 0–3%, and 0–1% for *lc*, *hc*, and *s*, respectively), we investigated the occurrence of positively charged residues R and K in the FG-segments and found that the *lc* group contains only 2–3% of positively charged residues in contrast to the *hc* and *s* groups, which have more positively charged residues (7–16% for *hc* and 13–17% for *s*, see Appendix A). Thus, it seems that the larger amount of positive charge in *hc* and *s* is more efficient in screening the effect of the negative charge increase induced from phosphorylation than the small amount of positive charges in the *lc* group (see Appendix A). In order to confirm this, we plotted the ratio of $\Delta R_{S}^{}/\left(N-1\right)b$ normalized by the fraction of phosphorylatable residues (*n*) as a function of the percentage of positive charge content (*p*) in the FG-segments (see Figure 2c). The data points in Figure 2c can be fitted to a straight line with a slope of -0.41 and *y*-intercept of 0.1, with an R^2^ value of 0.68. Using this observation, the Stokes radius for a phosphorylated FG-segment can be predicted using the following expression:
(1)$$R_{S}^{\mathrm{phos}}=R_{S}^{\mathrm{native}}+bn\;\left(N-1\right)\left(-0.41p+0.1\right)$$
We show the predictive power of this formula in Figure 2d, where the $R_{S}^{\mathrm{phos}}$ predicted using Equation (1) is plotted against the computed $R_{S}^{\mathrm{phos}}$ from the MD simulations, showing a very good correlation (with *R*^2^ = 0.97). 

### 2.2. Effect of Phosphorylation on NPC Structure

For the next step, we analyzed the disordered protein distribution inside the yeast NPC upon phosphorylation. Due to phosphorylation according to the Phos_N scheme, 6432 (7.4%) of all residues (86520) are phosphorylated inside the NPC, whereas for Phos_Max, 20992 (24.3%) of the NPC residues are phosphorylated (see Appendix A). We tethered the FG-Nups inside the yeast scaffold at the same anchoring points as in the wild type yeast NPC [35,48], and then switched on phosphorylation. In Figure 3, we show snapshots of the wild type and phosphorylated NPCs (for both the Phos_N and Phos_Max conditions). As expected from the conformational analysis of isolated FG-Nups (Section 2.1), the FG-Nups assumed extended conformations in the phosphorylated NPC compared to the wild type NPC. For the maximally phosphorylated (Phos_Max) state, the FG-Nups spilled out of the NPC over a distance of almost 100 nm from the NPC surface, which is considerably larger than that for the Phos_N state. In the case of isolated FG-Nups, only intra-molecular interactions are present, whereas inside the NPC, the residues of each FG-Nup also interact with residues from other FG-Nups (i.e., both intra- and inter-molecular interactions are present). Therefore, the enhanced electrostatic repulsion and reduced hydrophobic attraction due to phosphorylation is much more pronounced inside a confined space like the NPC, resulting in a strong reduction of protein density.

We analyzed the collective distribution of the FG Nups inside the NPC by calculating the time averaged radial density distribution (averaged over the axial and circumferential direction) inside the pore (i.e., for |*z*| < 15.5 nm) (note that the origin of the coordinate system coincides with the center of the NPC). The radial density profile for all residues and hydrophobic residues are plotted in Figure 4a,b, respectively, for the wild type and phosphorylated NPCs. In addition, the 2-dimensional (*rz*) density distribution (averaged over the circumferential direction) is plotted in Figure 4c for the wild type and phosphorylated NPCs. For technical details on calculating the density distributions, the reader is referred to the Materials and Methods section. Figure 4 clearly shows that phosphorylation significantly alters the density distribution of the FG-Nups inside the NPC. For a wild type NPC, the mean density at the center (0 nm < *r* < 5 nm) is ~ 80 mg/mL (see Figure 4a), which gradually increases with the radial distance *r* from the center and attains a peak value (~180 mg/mL) at *r* ~ 15 nm, after which the density gradually decreases. This is consistent with the doughnut-like structure in Figure 4c (left panel). However, for the phosphorylated NPCs, the density drops drastically inside the pore, amounting to only ~50 mg/mL and ~20 mg/mL at the center (see Figure 4a) for the Phos_N and Phos_Max scenarios, respectively. In the case of Phos_N, the radial density profile follows a similar trend (but lower in magnitude) as the wild type. This results in a less dense doughnut-like structure, as shown in Figure 4c (middle panel). For Phos_Max, the density remains lower than 20 mg/mL throughout the full range of *r*-values, as shown in Figure 4a,c (right panel). The density distribution of hydrophobic residues (see Figure 4b) is highly correlated with the density distribution of the total amount of residues (see Figure 4a). 

To explore the main reason for the reduction in protein density for both phosphorylated NPCs, we computed the relative contribution of the hydrophobic and electrostatic energy to the total interaction energy inside the wild-type and phosphorylated NPCs [15,34,35], as shown in Figure 5a,b, respectively. In the wild type NPC, the time-averaged hydrophobic interaction energy amounts to approximately −76,300 kJ/mol, whereas for the Phos_N and Phos_Max NPCs, these values are about −49,000 kJ/mol and −6100 kJ/mol, respectively. Here, by far the largest reduction is in the Phos_Max NPC, with almost a twelve-fold decrease in hydrophobic interaction energy relative to the wild type. Note that this twelve-fold decrease cannot be explained by the reduction in net hydrophobicity alone (a reduction of 16%, see Appendix A); also, the distance between the hydrophobic amino acids plays an important role, being much larger for Phos_Max than for the wildtype and Phos_N NPCs (see Figure 4). On the other hand, the Coulomb energy was measured to be two orders of magnitude smaller than the hydrophobic energy for the wild type and Phos_N NPCs (around −750 kJ/mol for wild type and −800 kJ/mol for Phos_N), while for Phos_Max, the total repulsive Coulomb energy (around 31200 kJ/mol) is much larger than the hydrophobic energy. In summary, the wild type and Phos_N NPCs are highly hydrophobic, with only a small contribution from electrostatics. In sharp contrast to this, the energy in the Phos_Max NPC has a much more dominant (repulsive) Coulombic contribution corresponding to the large net negative charge, while the hydrophobic energy is much lower.

### 2.3. Effect of Phosphorylation on Active and Passive Transport

In this section, we focus on the selective permeability barrier of the NPC and how phosphorylation affects this. The nuclear transport receptor Kap95 is known to interact with the FG-Nups via its hydrophobic binding sites and to translocate through the NPC by facilitated transport in the presence of RanGTP at the nucleoplasmic side, which dissociates the Kaps from the NPC [49,50]. Beside being hydrophobic, the Kaps are also negatively charged [51]. It is therefore expected that the interaction between the Kaps and FG-Nups is strongly affected by the phosphorylation-induced charge modification and reduction in hydrophobicity of the FG-Nups. To investigate this, we modelled a yeast NPC in the presence of ten Kap95 particles (with a diameter of 8.5 nm, 10 hydrophobic binding sites, and a uniformly-distributed surface charge of −43e, as used previously [15]) that are released at the cytoplasmic side to probe facilitated transport. After equilibrating the system (see Materials and Methods for details), the simulations were carried out for the same initial positions of the Kaps for the wild type and phosphorylated pores. Snapshots of the final state (at *t* = 2 μs) are shown in Figure 6, illustrating the inhibition of facilitated transport in phosphorylated NPCs. The bottom panels of Figure 6a–c depict a reduced binding affinity of the Kaps with the phosphorylated FG-Nups. To assess the propensity for translocation, we plot the initial (*t* = 0 μs) and final (*t* = 2 μs) *z* coordinate of the center of mass of the Kap95 particles in Figure 7a. We observed that for the wild type NPC, the Kap95-FG-Nup affinity is larger compared to the phosphorylated NPCs (see Figure 6), so that the Kap95 particles are able to enter the pore and translocate (Figure 7a). The large affinity is due to the fact that the pore is hydrophobic (see Figure 5 and Appendix A) and has a weak positive charge [35]. In the course of 2 μs, a total of 9 Kaps translocated through the pore and the remaining Kap ended up inside the pore (Figure 6a and Figure 7a). In the phosphorylated NPC, however, despite the lower FG-Nup density (see Figure 4), the Kaps are excluded from the pore (Figure 6b,c and Figure 7a). The results from Figure 5, Figure 6, and Figure 7a point towards a phosphorylation-induced decrease in hydrophobicity (for Phos_Max and Phos_N) and increase in coulombic repulsion (for the Phos_Max case) resulting in lowering of Kap95-FG-Nup binding affinity, which is instrumental for active transport.

Next, we study the effect of phosphorylation on passive transport by probing the transport of inert particles of the same size as the Kap95 particles (i.e., 8.5 nm in diameter) but without charge and hydrophobic binding spots. We used the same initial positions for the inert particles as for the Kap95 particles used in the case of active transport (Figure 6 and Figure 7a). In Figure 7b, we plot the initial and final *z* location of the center of mass of the inert particles for the wild type and phosphorylated NPCs. For the wild type NPCs, it can be clearly observed that the inert particles stay at the cytoplasmic side and do not enter the NPC. On the other hand, in the Phos_Max NPC, two inert particles managed to translocate through the pore within 2 µs, showing that the permeability barrier of the Phos_Max NPC is jeopardized. For the Phos_N NPC, we did not observe any translocation. To further test the size-dependent permeability barrier of the Phos_N and wild type NPCs, we performed two additional transport simulations for ten spherical inert particles of diameter 4 nm. The results are shown in Appendix A (see the Appendix A), revealing that in the wild type, the total number of translocations is 129, whereas in the Phos_N pore, there were 328 translocation events, indicating a 2.5-fold increase in passive transport rate compared to wild type. For the wild type, this is consistent with our previous work [33], where we found the energy barrier for inert particles of size 4 nm to be lower than $\kappa_{B}T$ (and thus likely to go through), while that for inert particles of 7 nm (and up) was found to be larger than 2 $\kappa_{B}T$ (and thus likely to not pass through). Our results for the wild type can be understood by recourse to the scaling relation of Timney et al. [8], which states that the characteristic time constant of passive transport scales with the third power of the molecular mass. Since the two inert particles used here are of size 4 and 8.5 nm, the mass dependence rule predicts that the translocation of the 8.5 nm particle should be 883 times slower compared to the smaller 4 nm particle. This is in qualitative agreement with our simulations, where for the 4 nm particles we see 129 translocation events within 2 μs of simulation time, whereas no translocation is observed for the 8.5 nm inert particle. Since Phos_Max NPCs have a lower protein density compared to Phos_N NPCs, we can expect the transport rates to be even higher for a 4 nm particle. In the case of passive transport, the inert particles interact with the FG-nups by means of steric repulsion only, and therefore the translocation events (Figure 7b and Appendix A) of inert molecules can be understood in terms of the density distribution of FG-Nups in the wild type and phosphorylated NPCs. Figure 4 shows that the FG-Nup density inside the NPC is significantly higher for the wild type NPC compared to the phosphorylated NPCs, which explains why in the wild type NPC no passive transport is observed (Figure 7b), whereas in the phosphorylated NPCs, passive transport occurs due to the lower permeability barrier.

Finally, we summarize our findings on active (from Figure 7a) and passive transport (Figure 7b and Appendix A) for the wild type, Phos_N, and Phos_Max NPCs in Figure 8. The wild type NPC is seen to have a selective permeability barrier, as it allows Kaps and small inert particles (diameter = 4 nm) to pass through, whereas larger inert particles (diameter = 8.5 nm) are excluded. The phosphorylated Phos_N NPC loses its selectivity, as transport of Kap95 is not observed, but its permeability barrier is still intact (8.5 nm particles are excluded). This indicates that the reduced steric hindrance due to the reduced amino acid density in the center (Figure 4a) is still sufficient to exclude large particles, but that the reduced hydrophobicity is no longer able to attract Kap95 particles into the FG-Nup mesh-work. Finally, the heavily phosphorylated Phos_Max NPC is observed to lose both its ability to facilitate active transport (due to the reduced hydrophobic attraction and increased electrostatic repulsion with respect to Kap95 particles), as well as its permeability barrier (due to the drastically reduced amino acid density allowing for transport of both inert particles). This, of course, is subject to the constraint of the limited time frame of our simulations (i.e., 2 μs). 

## 3. Discussion 

In the current study, we incorporated phosphorylation-induced modifications of the hydrophobicity and charge of the four amino acids S, H, Y, and T (see Table 1) into our 1BPA model for IDPs. We addressed the effect of phosphorylation on the conformational changes of 16 different isolated FG-Nup segments of varying length and with varying numbers of charged and hydrophobic residues. We compared the predicted Stokes radius $R_{S}^{}$ of these FG-Nup segments in their phosphorylated state with the values in their native state (see Figure 1), and observed an increase in size due to phosphorylation. We found that $R_{S}^{}$ increases linearly with the fraction of phosphorylatable residues and decreases linearly with the percentage of positively-charged amino acids (see Equation (1) and Figure 2d. While the former dependence is straight-forward, the latter is subtler and points to the important role of positive charge in screening the effect of the phosphorylation-induced increase in negative charge. 

Next, we investigated how the FG-Nups interact when confined inside the NPC and how phosphorylation alters these interactions and the resulting protein distribution. The density distributions demonstrate a considerable difference between the wild type and phosphorylated NPCs, with the total amino acid density and hydrophobic density dropping by almost a factor two and four for the Phos_N and Phos_Max NPCs, respectively, in comparison to wild type. Whereas the hydrophobicity changed both in terms of density (Figure 4b) and energy (Figure 5a) for the two phosphorylated NPCs, only the Phos_Max NPC showed a large increase in (repulsive) electrostatic energy, while for the Phos_N and wild type NPC, the electrostatic energy remained negligible compared to the hydrophobic energy (Figure 5a,b). All considered, we can conclude that the phosphorylated FG-nups resulted in a higher negative charge and lower hydrophobicity, resulting in a strong depletion of amino acid density in phosphorylated NPCs, with the effects (especially the electrostatic) much more pronounced in Phos_Max NPCs.

For those molecules that translocate through the NPC by means of active transport (for example Kap95 in this study) [15,33,34,35], the molecular interactions can be divided into three components: (i) steric repulsion by means of excluded volume; (ii) hydrophobic interactions; and (iii) Coulombic interactions. Firstly, as the density inside the phosphorylated pores (both Phos_N and Phos_Max NPCs) is significantly lower than in the wild type (see Figure 4), the steric repulsion component is lower. Secondly, as illustrated in Figure 5a, phosphorylation results in a serious reduction of the hydrophobic interaction energy, as the residues become more hydrophilic upon phosphorylation. Finally, as Kap95 carries negative charge, it will face electrostatic attraction when the pore is positively charged (wild type) and electrostatic repulsion when the pore is negatively charged (phosphorylated). All the energy components taken together indicate that the negatively-charged and hydrophobic Kap95 experiences a much more repulsive environment inside the phosphorylated NPCs due to the increased negative charge and reduced hydrophobicity compared to wild type. Therefore, the overall energy barrier for the translocation of Kap95 particles through a phosphorylated pore is much higher compared to the wild type pore. Our transport simulations (Figure 6 and Figure 7a) for Kaps indeed reveal inhibition of facilitated transport upon phosphorylation, while wild type NPCs facilitate Kap translocation. We do not account for the presence of RanGTP in our model, which is known to play an important role in dissociating the Kaps from the FG-Nups. As a result, the model Kaps remain in the bound state towards the nuclear side of the pore thanks to a slightly higher affinity of the nuclear FG-Nups with the Kap95 [52]. In contrast to this, the phosphorylated pore inhibits the Kaps to enter the pore. Of course, we cannot completely rule out the fact that some of the Kaps might translocate through phosphorylated NPCs at longer simulation times. Nevertheless, the trend of a reduced probability for active transport through phosphorylated pores and an increased probability for passive transport upon phosphorylation is in accordance with experimental observations [27,29,30,31,32]. 

The results for wild type and Phos_N are comparable to our previous studies on biomimetic nanopores coated with Nsp1 and a more hydrophilic mutant, Nsp1-S, which illustrated that a lack of cohesion in the hydrophilic Nsp1-S pore can result in a depleted density (~twofold decrease) compared to the hydrophobic Nsp1 pore [15,53]. Despite the different nature of the modification (mutation versus phosphorylation), both cases result in a reduced protein density due to a depleted hydrophobicity, while the electrostatic interactions remain approximately unaffected (see Figure 5). However, the effect on the selective permeability was found to be different. Whereas the Nsp1-S pore lost its selectivity due to the fact that the permeability barrier was jeopardized (large inert particles were found to go through), the Phos_N NPC retained its permeability barrier, but lost its ability to actively transport Kap95 particles. This difference is most likely related to the different unfolded protein composition of the yeast NPC and the biomimetic nanopore, with the former consisting of 10 different FG-Nups, while the latter has only one.

It should be noted that in this study, we compared the wild type results with an NPC in which all S, H, T, and Y residues are maximally phosphorylated (Phos_Max), and a second variant (Phos_N) in which the phosphorylation sites are extracted from the NetPhosYeast 1.0 server [37], accounting for the phosphorylation of a subset of all S and T residues. Clearly, the Phos_Max scenario is not very relevant from a biology point of view, as phosphorylation of all S, T, H, and Y does not occur simultaneously in reality. The results of the Phos_Max scenario therefore serve as a theoretical limiting case of phosphorylated NPCs that feature a maximal phosphorylation-induced modification of charge and hydrophobicity. The Phos_N scenario predicts a higher number of phosphorylation sites compared to other phosphorylation databases, such as the fungi phosphorylation database (FPD) (see Appendix A for FPD phosphosites). Despite this difference, for both scenarios we observe loss of selectivity (see Figure 7a and Appendix A), while the permeability barrier is retained (see the section “Sensitivity analysis” in the Appendix A).

It is still not known what fraction of the phosphorylatable residues predicted in these databases actually undergo simultaneous phosphorylation inside the NPC in vivo. This will be an interesting aspect to be explored further, since our study indicates that the degree of phosphorylation can have a large impact on the structure of the NPC and the rate of transportation in passive and active pathways. Our work should therefore not be seen as an exact mimic of specific biological conditions, but as a qualitative study, in which NPC phosphorylation is explored in order to shed light on the fundamental mechanisms underlying in vitro experiments on the decreased kinetics for active import [25,27,29] and the increased kinetics of passive import [32] in phosphorylated NPCs. 

## 4. Materials and Methods 

### 4.1. Coarse-Grained Molecular Dynamics Simulations

The 1BPA Molecular Dynamics model used in this study accounts for the exact amino acid sequence of the FG-Nups, in which each bead is located at the *C_α_* positions of the polypeptide chain [35,46]. We set the mass of each bead to the average amino acid mass (120 Da), and the distance between neighboring beads to ~0.38 nm through a stiff harmonic spring potential. The bending and torsion potentials are extracted from the Ramachandran data of the coiled regions of protein structures [46]. The solvent molecules are treated in an implicit manner. A distance-dependent dielectric constant is used to account for the solvent polarity, and ionic screening is incorporated through Debye screening with a screening constant *k* = 1 nm^−1^ corresponding to the physiological salt concentration inside the NPC [54]. The hydrophobic interactions between the amino acids are incorporated through a modified Lennard-Jones potential, which accounts for hydrophobicity scales of all 20 amino acids derived from normalized experimental partition energy data renormalized in a range from 0 to 1. For details of the method, the reader is referred to [35].

All MD simulations were carried out with a time step of 0.02 ps [35]. The simulations for the isolated disordered FG-Nup segments were carried out for 2.5 × 10^7^ steps [35], which was found to be sufficiently long to reach convergence. For the NPC simulations with particles (Figure 6, Figure 7 and Appendix A) and without particles (Figure 3, Figure 4, and Figure 5), the systems were first energy minimized to remove any overlap of the amino acid beads. Then, all long-range forces were gradually switched on, and for the NPC with particle systems, the inert/Kap95 particles were kept at a fixed position on the cytoplasmic side. In the final production run for the NPC without particles, the simulations were carried out for 5 × 10^7^ steps (with the first 5 × 10^6^ steps ignored so that only the statistically meaningful results are extracted), which was found to be long enough to have converged results for the density distribution inside the pores. For the NPC with particles, we included one additional step before the production runs, in which we equilibrated the system for 5 × 10^6^ steps with all long-range forces switched on while keeping the inert/Kap95 particles fixed at their position. In the final production runs for the NPC with transporting particles, the inert/Kap95 particles were allowed to move and the simulations were carried out for 10^8^ steps. For the Kap95 simulations we modelled the hydrophobic binding sites on the Kaps as F beads [15,33].

The time-averaged density calculations presented in the main text (see Figure 4) were derived by using the “gmx densmap” tool in GROMACS. The nanopore is centered inside a box of size 100 nm × 100 nm × 200 nm, which was divided into discrete cells of size 0.5 nm × 0.5 nm × 0.5 nm. The trajectory files from the simulations were analyzed to compute the number density in each cell as a function of simulation time. A time averaged 3D mass density profile was obtained by multiplying the number density with the mass of each bead and then averaging over the simulation time. The 3D density was averaged in the circumferential direction to obtain two-dimensional (2D) *rz* density plots (as shown in Figure 4c). Finally, the radial density distribution was obtained by averaging these 2D density maps in the vertical direction (as shown in Figure 4a,b). To compute the Coulombic and hydrophobic interaction inside the NPC (see Figure 5), we used the “gmx energy” tool from GROMACS.

### 4.2. Parametrization of Phosphorylated Amino Acids

We used five different hydrophobicity-predictor programs to estimate the hydrophobicity of phosphorylated residues. These programs calculate the logarithmic value of the equilibrium partition coefficient *P*, i.e., the ratio of concentrations in a mixture of two immiscible phases, water, and 1-octanol, as a measure for hydrophobicity. They use experimental log *P* values of fragments (small groups of atoms) to calculate the log *P* for bigger molecules by adding the individual contributions from the constituting fragments, based on the structure additivity principle for hydrophobicity [55]. There are several challenges in incorporating the log *P* estimates for the complete molecules directly in our 1 BPA model, which are: (i) the error from the estimate of the log *P* values for the fragments accumulate while calculating the log *P* for the entire molecule; and (ii) each of the hydrophobicity-predictor programs are trained with different experimental data, and therefore generate different estimates of log *P* for a given molecular structure. In order to have the hydrophobicity values comparable to our 1BPA model, first we rescaled and then normalized the log *P* estimates for all amino acids obtained from a hydrophobicity-predictor program *k*, so that hydrophobicity of any amino acid *i* (i.e., $\varepsilon_{k,i}$) falls in the range from 0 to 1. Here, 0 and 1 corresponds to the hydrophobicity of the most hydrophilic and most hydrophobic amino acids, according to hydrophobicity-predictor program *k*. Next, to minimize the error we decided to incorporate the change in $\varepsilon_{k,i}$ values of the phosphorylatable residues (obtained from the five hydrophobicity-predictor programs) compared to the 1BPA model, for which the hydrophobicity values are extracted from three different partition coefficient measurements [35]. To account for the variation in the prediction of $\varepsilon_{k,i}$ by the different hydrophobicity-predictor programs, a weighted average approach is considered. The weights are assigned to individual predictor programs based on their accuracy in predicting the $\varepsilon_{k,i}$ values of the amino acids in their native state, as used in [35]. Thus, the assigned weight for hydrophobicity-predictor program *k* for amino acid *i* can be written as,
(2)$$w_{k,i}=\frac{{\left(1/\Delta \varepsilon_{k,i}\right)}^{2}}{\sum_{k=1}^{5}{\left(1/\Delta \varepsilon_{k,i}\right)}^{2}},$$
where $\Delta \varepsilon_{k,i}=\varepsilon_{k,i}-\varepsilon_{1\mathrm{BPA},i}$ represents the difference between the hydrophobicity for an amino acid in its native state used in our 1BPA model [35] and the hydrophobicity-predictor programs (see Appendix A for the source data). Next, the change in hydrophobicity upon phosphorylation is calculated as $\Delta \varepsilon_{k,i\text{-}\mathrm{phos}}=\varepsilon_{k,i\text{-}\mathrm{phos}}-\varepsilon_{1\mathrm{BPA},i}$, where “*i*-phos” represents the amino acid *i* in its phosphorylated state. Finally, using the weights for the hydrophobicity-predictor programs (see Appendix A) we computed the hydrophobicity for the phosphorylated amino acid as $\varepsilon_{p,i}=\varepsilon_{1\mathrm{BPA},i}+$$\sum_{k=1}^{5}w_{k,i}\;\Delta \varepsilon_{k,i\text{-}\mathrm{phos}}$. The amino acids Serine (S), Histidine (H), Tyrosine (Y), and Threonine (T) undergo phosphorylation [18,19], and the introduction of a phosphate group results in the introduction of a −2e charge, as shown in Table 1. The phosphorylation of these amino acids results in a more hydrophilic atomic composition, which can be seen in Table 1. As a reference, the prediction of the hydrophobicity of amino acid *i* in the native state, $\varepsilon_{\mathrm{weighted},i}=\varepsilon_{1\mathrm{BPA},i}+\sum_{k=1}^{5}w_{k,i}\;\Delta \varepsilon_{k,i}$, is also shown in Table 1. Note that the subscript *i* is dropped from $\varepsilon_{p,i}$ and $\varepsilon_{\mathrm{weighted},i}$ in Table 1 for clarity.

## Figures and Tables

**Figure 1 ijms-20-00596-f001:**
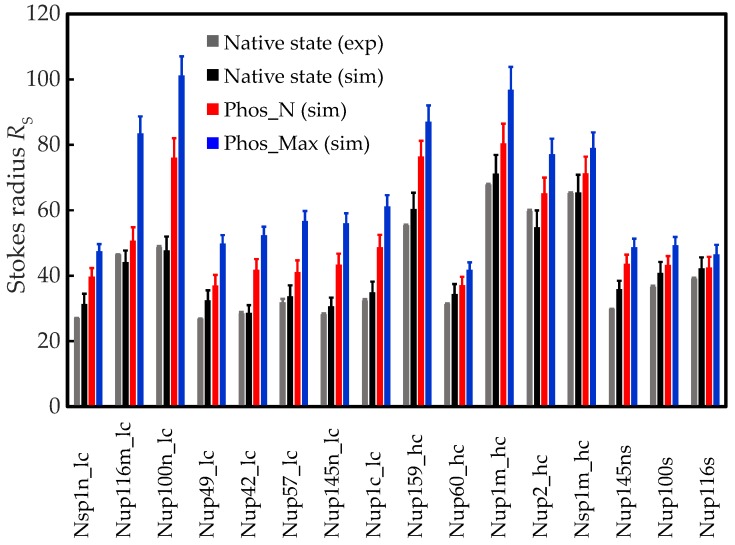
Phosphorylation-induced extension of FG-Nup segments. The Stokes radius $R_{S}^{}$ (in Angstrom) is depicted for a range of FG-Nup segments in their native and phosphorylated states. The suffix *lc* denotes low charge, *hc* high charge and *s* refers to the stalk region of the Nup. The grey and black bars represent the data in the native state from experiments [39] and simulations (results reproduced from [35]), respectively, and the prediction for the phosphorylated states are plotted in red for Phos_N and blue for Phos_Max. For the simulation data, the error bars represent the standard deviation in time (see Appendix A for the source data).

**Figure 2 ijms-20-00596-f002:**
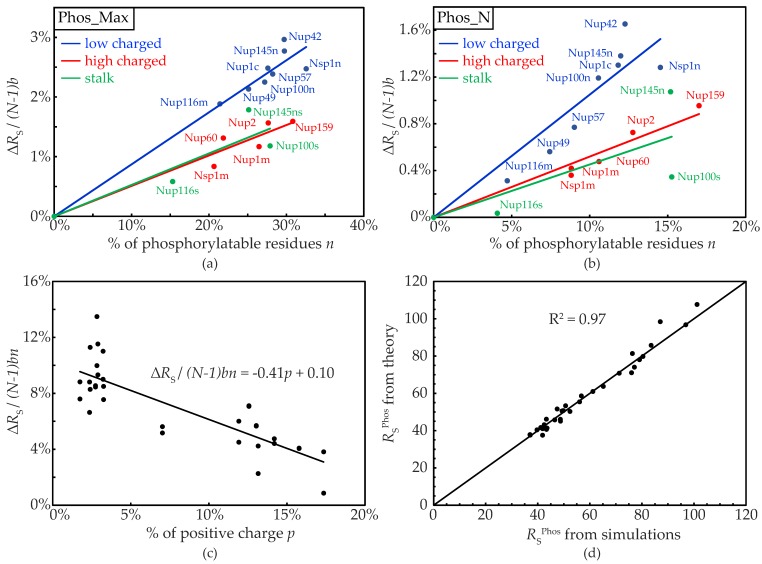
The normalized change in $R_{S}^{}$ (i.e., $\Delta R_{S}^{}/\left(N-1\right)b)$ due to phosphorylation as a function of the fraction of phosphorylatable residues (*n*) for (**a**) the Phos_Max and (**b**) the Phos_N scenarios [37]. The expression for the change in $R_{S}^{}$ (i.e., $\Delta R_{S}^{}=R_{S}^{\mathrm{phos}}-R_{S}^{\mathrm{native}}$), with $R_{S}^{\mathrm{phos}}$ and $R_{S}^{\mathrm{native}}$ being the Stokes radii of the FG-Nup segments in the phosphorylated and native states, respectively, is normalized with (*N* − 1)*b* where *N* is the total number of residues of the FG-Nup segment, and *b* (= 3.8 Angstrom) is the coarse-grained bond length between neighboring amino acids [35,46]. The data for the FG-Nups from the high charged (*hc*), low charged (*lc*), and stalk (*s*) segments [35,39] are represented in red, blue, and green data points, respectively. The data points of each group are fitted to a straight line passing through the origin revealing different slopes for different groups. For Phos_Max we observe slopes of 0.087 for *lc* (*R*^2^ = 0.94), 0.051 for *hc* (*R*^2^ = 0.92), and 0.053 for *s* (*R*^2^ = 0.80), respectively, whereas for Phos_N the slopes are 0.1 for *lc* (*R*^2^ = 0.86), 0.052 for *hc* (*R*^2^ = 0.95) and 0.045 for *s* (*R*^2^ = 0.61). (**c**) The ratio of normalized change in $R_{S}^{}$ to the fraction of phosphorylatable residues (*n*) is plotted as a function of the fraction of positively charged residues (*p*) for all data points (black) from the Phos_Max and Phos_N scenarios. These data points are fitted to a linear equation, as shown in the figure (giving *R*^2^ = 0.68). (**d**) The $R_{S}^{\mathrm{phos}}$ predicted from the theory in Equation (1) compared to $R_{S}^{\mathrm{phos}}$ computed from the MD simulations, both in Angstrom, show a good agreement with a fitness measure of *R*^2^ = 0.97.

**Figure 3 ijms-20-00596-f003:**
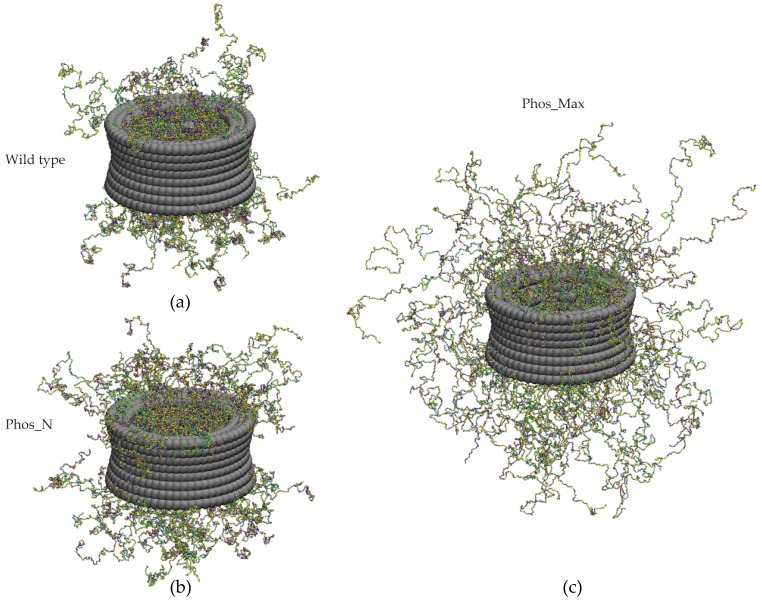
Snapshots of the coarse-grained MD models of (**a**) the wild type NPC, (**b**) the Phos_N NPC, and (**c**) the Phos_Max NPC. The FG-Nups (different color beads denote different amino acids) are attached at the same anchor positions, as in [35,48]. Phosphorylation-induced changes in the interaction leads to spilling of FG-Nups out of the NPC.

**Figure 4 ijms-20-00596-f004:**
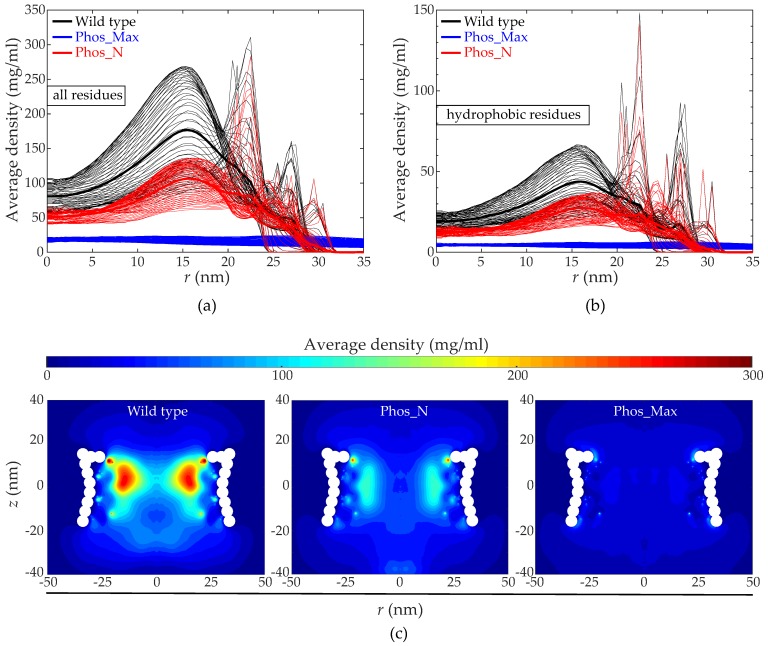
Disordered protein structure inside the wild-type and phosphorylated NPCs. (**a**,**b**) Time-averaged radial density distribution inside wild type (black), Phos_N (red) and Phos_Max (blue) NPCs for (**a**) all residues, and (**b**) hydrophobic residues. The thin lines represent the density at different positions along the *z*-axis separated by 1 nm, in the range of |*z*| < 15.5 nm (height of the NPC), with the mean density plotted as thick lines. (**c**) The *rz*-density map for the wild type (left panel), Phos_N (middle panel), and Phos_Max (right panel) NPCs. The wild type shows the characteristic highly dense doughnut-like structure [35]. The Phos_N NPC shows a density-depleted doughnut-like structure, whereas the Phos_Max NPC shows a significantly less dense and rather uniform density distribution.

**Figure 5 ijms-20-00596-f005:**
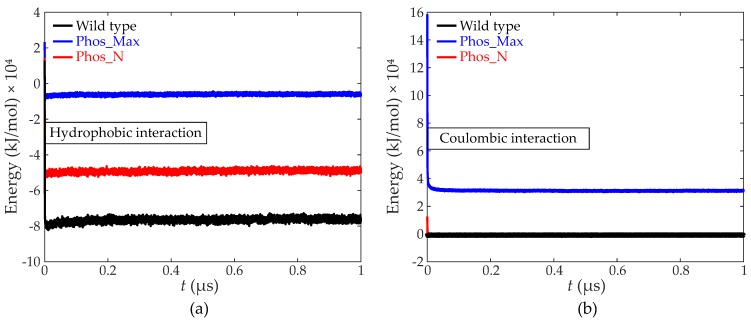
Time evolution of (**a**) the hydrophobic interaction energy, and (**b**) the coulombic interaction energy of the FG-Nups for the wild type (black), Phos_Max (blue), and Phos_N (red) NPCs.

**Figure 6 ijms-20-00596-f006:**
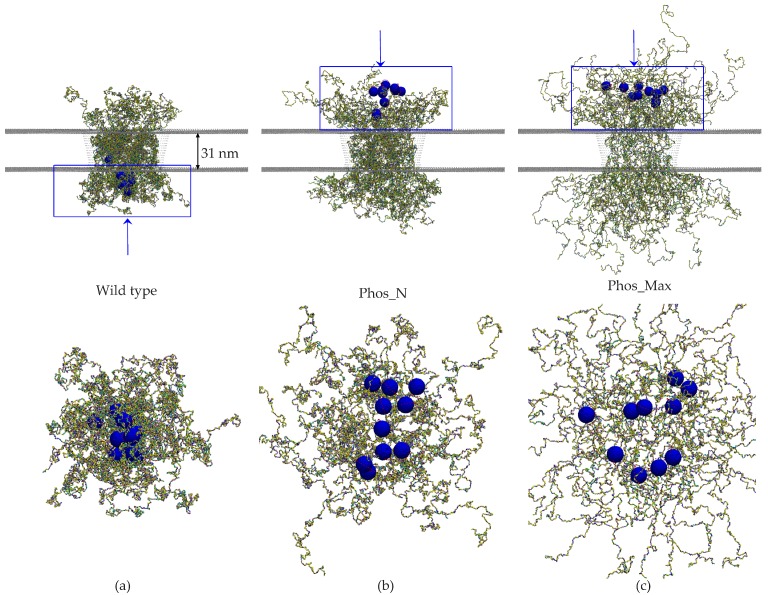
Snapshots at *t* = 2 μs of the FG-Nups and model Kap95 particles inside a wild type NPC (**a**), a phosphorylated NPC according to the Phos_N scheme (**b**), and a phosphorylated NPC according to the Phos_Max scheme (**c**). The Kap95 particles are shown in blue with the red hydrophobic binding spots on its surface. The 20 different amino acids of the FG-Nups are represented by different colors. The size of the scaffold beads (grey) is scaled down to make the Kap particles better visible. In the bottom panel we provide a magnified view from the bottom for (**a**) and from the top for (**b**) and (**c**), focusing on the region indicated by the blue boxes in the top panels. The Kaps are shown to traverse the wild type NPC, whereas the Kaps are not able to strongly partition into the FG-Nup meshwork for the phosphorylated NPCs.

**Figure 7 ijms-20-00596-f007:**
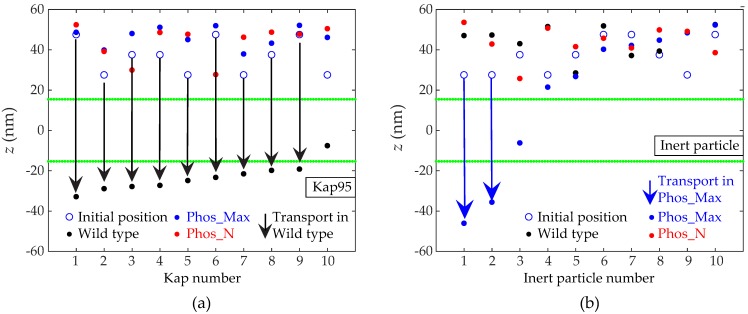
Effect of phosphorylation on active (**a**) and passive (**b**) transport. In both cases, the particles are released from the same position at the cytoplasmic side and are ordered from left to right based on the end position at *t* = 2 μs. (**a**) Initial (*t* = 0 μs) and final (*t* = 2 μs) axial (*z*) position of ten Kap95 particles, and (**b**) Initial (*t* = 0 μs) and final (*t* = 2 μs) axial (*z*) position of ten inert particles of the same size as the Kap95 particles (diameter = 8.5 nm, no charge and no hydrophobic binding spots). In both (**a**) and (**b**), the boundaries of the NPCs (|*z*| = 15.5 nm) are represented by green lines and the arrows represent translocations from the cytoplasm to the nucleoplasm.

**Figure 8 ijms-20-00596-f008:**
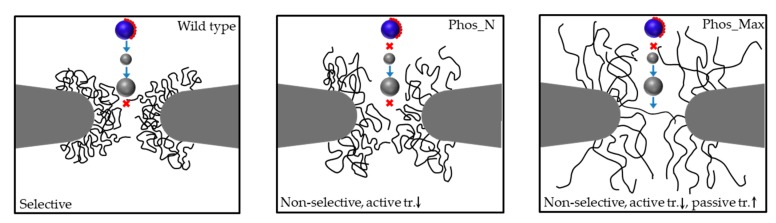
Summary of simulation results on wildtype and phosphorylated yeast NPCs, showing active transport of Kap95 (blue sphere with red dots representing the hydrophobic binding sites) and passive transport of inert particles (grey spheres of diameter 4 nm and 8.5 nm). The FG-Nups are represented by black filaments. The blue arrows indicate observed transport and the red crosses indicate prohibited transport within the 2 μs simulation time.

**Table 1 ijms-20-00596-t001:** Parameters in the 1BPA forcefield for phosphorylated amino acids. Note that: $\varepsilon_{1\mathrm{BPA}}$ and $\varepsilon_{\mathrm{weighted}}$ are the normalized hydrophobicity values (between 0 and 1) from the 1BPA model [35] and the weighted average scheme (see Section 4) for the amino acids in their native state, respectively; $\varepsilon_{p}$ is the hydrophobicity of the phosphorylated amino acid; and q and $q_{p}$ denote the charge of the amino acids in their native and phosphorylated conditions, respectively.

AA	*ε* _1BPA_	*ε* _weighted_	*ε* _p_	*q*	*q* _p_
Ser (S)	0.45	0.41	0.07	0	−2e
His (H)	0.53	0.44	0.06	0	−2e
Thr (T)	0.51	0.52	0.23	0	−2e
Tyr (Y)	0.82	0.83	0.67	0	−2e

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
