# Peer review of "The Effect of FG-Nup Phosphorylation on NPC Selectivity: A One-Bead-Per-Amino-Acid Molecular Dynamics Study"

_ijms, 2019, doi:10.3390/ijms20030596_

Reviewer 1 Report

The manuscript by Mishra and coworkers describes an in silico study, based on the so-called 1-bead-per-amino-acid molecular dynamics approach, of the change in properties due to massive phosphorylation events of proteins located in the inner wall of the nuclear pore complexes (NPC). These are intrinsically disordered proteins rich in FG-repeats, for which some experimental evidence has been gathered in that phosphorylation leads to decreased kinetics of active transport of certain nuclear transport receptors (Kap95) whereas results in increased kinetics of passive transport. In line with this previous evidence, the authors reach the conclusion that phosphorylation causes a reduced attraction between the residues of nucleoporins, leading to a less dense FG-network inside the NPC. Furthermore, they show by mean of computer simulation that upon phosphorylation the transport rate of nuclear transport receptors decreases whereas that of inert molecules increases and propose that these changes can be explained by modified hydrophobic, electrostatic and steric interactions.

Modulation of fluxes through the NPC is an important aspect of the cell biology, determining many key events in the cell. In this regard, the study is worth considering. The paper is concise and correctly written. My major concern derives from the doubt that the predicted effects could have any real physiological meaning. First of all, there is no description about how the predicted phosphorylation residues obtained from the NetPhosYeast (Phos_N scenario) have been filtered. Even if the default threshold has been used (score 0.5), this is a very conservative setting, and the “surviving” phosphosites in many cases are not real sites. In my opinion, a study in which ALL putative phosphorylatable Ser and Thr sites ARE SIMULTANEOUSLY phosphorylated is a situation extremely unlikely and any conclusion drawn from such situation has dubious biological relevance. Just as an example, while Table s2 calculates 15% of phosphorylatable residues for Nsp1n_lc, according SGD none of these residues have been reported to be phosphorylated in vivo.

Even more unlikely is the Phos_Max scenario in which, in addition, ALL His and Tyr are also phosphorylated. Now, phosphorylation in Tyr is a very rare event in yeast, and so is the presence of P-His, often restricted to a fungal variation of the two-component system found in prokaryotes. I wonder why the authors have not considered a more physiological scenario, computing the effects of phosphorylation at known sites found experimentally, taking the data from SGD (i.e. via YeastMine) or from the even more complete FPD database generated by Bai and coworkers (Fungal Biol. 2017 121:869-875. doi: 10.1016/j.funbio.2017.06.004).

Author Response

Please see the

Reviewer 1:

The manuscript by Mishra and coworkers describes an in silico study, based on the so-called 1-bead-per-amino-acid molecular dynamics approach, of the change in properties due to massive phosphorylation events of proteins located in the inner wall of the nuclear pore complexes (NPC). These are intrinsically disordered proteins rich in FG-repeats, for which some experimental evidence has been gathered in that phosphorylation leads to decreased kinetics of active transport of certain nuclear transport receptors (Kap95) whereas results in increased kinetics of passive transport. In line with this previous evidence, the authors reach the conclusion that phosphorylation causes a reduced attraction between the residues of nucleoporins, leading to a less dense FG-network inside the NPC. Furthermore, they show by mean of computer simulation that upon phosphorylation the transport rate of nuclear transport receptors decreases whereas that of inert molecules increases and propose that these changes can be explained by modified hydrophobic, electrostatic and steric interactions.

Modulation of fluxes through the NPC is an important aspect of the cell biology, determining many key events in the cell. In this regard, the study is worth considering. The paper is concise and correctly written. My major concern derives from the doubt that the predicted effects could have any real physiological meaning. First of all, there is no description about how the predicted phosphorylation residues obtained from the NetPhosYeast (Phos_N scenario) have been filtered. Even if the default threshold has been used (score 0.5), this is a very conservative setting, and the “surviving” phosphosites in many cases are not real sites. In my opinion, a study in which ALL putative phosphorylatable Ser and Thr sites ARE SIMULTANEOUSLY phosphorylated is a situation extremely unlikely and any conclusion drawn from such situation has dubious biological relevance. Just as an example, while Table s2 calculates 15% of phosphorylatable residues for Nsp1n_lc, according SGD none of these residues have been reported to be phosphorylated in vivo.

Even more unlikely is the Phos_Max scenario in which, in addition, ALL His and Tyr are also phosphorylated. Now, phosphorylation in Tyr is a very rare event in yeast, and so is the presence of P-His, often restricted to a fungal variation of the two-component system found in prokaryotes. I wonder why the authors have not considered a more physiological scenario, computing the effects of phosphorylation at known sites found experimentally, taking the data from SGD (i.e. via YeastMine) or from the even more complete FPD database generated by Bai and coworkers (Fungal Biol. 2017 121:869-875. doi: 10.1016/j.funbio.2017.06.004).

Response: We thank the reviewer for the in-depth review of our work, the constructive comments, and the positive recommendations. The major concerns raised by the reviewer are: (i) the Phos_Max scenario is not biologically relevant as phosphorylation of all S, T, H and Y does not occur in reality, (ii) Information is lacking regarding the threshold score in extracting the phosphorylation sites from NetPhosYeast (for the Phos_N scenario) and (iii) even with a conservative setting (score = 0.5) the Phos_N scenario can predict phosphosites which in many cases are not real sites and therefore the reviewer recommends to use experimentally verified phosphorylation sites such as the SGD or the even more complete FPD databases. In the following we provide a pointwise response to the above questions:

(i)     We agree with the reviewer’s remark on the Phos_Max scenario. However, the work on Phos_Max should be considered only to understand the transport mechanism of the NPC as a function of net charge and hydrophobicity as a theoretical limiting case of maximally phosphorylated NPCs. We have incorporated this in the second to last paragraph of the Discussion and Conclusion section in the revised manuscript to clarify this.

(ii)    We used the default setting of the NetPhosYeast1.0 server (threshold score = 0.5) for sites that can be phosphorylated. We have now mentioned this explicitly in the caption of Table s2 (line 39-40 of Supplementary Materials).

(iii)  We welcome this comment by the reviewer. To address this we have carried out additional simulations on NPCs that feature phosphorylated FG-Nups from the fungi phosphorylation database (FPD). The results are presented and discussed in the section ‘sensitivity analysis’ in the Supplementary Materials (see the new Tables s6 and s7 and Figure s2), showing that, despite the lower number of phosphosites in the FPD database, for both scenarios we observe loss of selectivity while the permeability barrier is retained. To further address this issue raised by the reviewer, we have modified the last paragraph of the Introduction and the second-to-last paragraph of the Discussion and Conclusions section.

Reviewer 2 Report

Authors revealed that extensive phosphorylation of FG-Nups decreases the selectivity of the NPC using one-bead per-amino-acid (1BPA) model. Overall, this study give us comprehensive insight how post-translational modification such as phosyphorylation is involved in traffic selectivity thorough NPC, and is interesting.

  However, (i) authors must include recent finding regarding phosphorylation of FG-NUPs and traffic selectivity. Last year, Hazawa et al. demonstrated that differentiation-inducible ROCK kinase phosphorylates on FG-domain of NUP62 and increased oncogenic TF p63 nuclear traffic in squamous cancer cell published in EMBO Reports (doi: 10.15252/embr.201744523.), which is further introduced in News and View (EMBO Reports, 2018).

In addition, (ii) it could be worthy to discuss phosphorylation cascade from the view of cell biology. Since it sounds too artificial if all of potential sites get phosphorylation together. Activity of kinase or phosphatase depend on cellular condition.

Finally, (iii) Are phosphorylation sites considered in this study conserved in mammalian cells? It would be better to be shown or discussed.

Author Response

Reviewer 2
Authors revealed that extensive phosphorylation of FG-Nups decreases the selectivity of the NPC using one-bead per-amino-acid (1BPA) model. Overall, this study give us comprehensive insight how post-translational modification such as phosyphorylation is involved in traffic selectivity thorough NPC, and is interesting.

 Response: We thank the reviewer for his in-depth review and interest in our work. We have responded pointwise below to the questions of the reviewer.

However, (i) authors must include recent finding regarding phosphorylation of FG-NUPs and traffic selectivity. Last year, Hazawa et al. demonstrated that differentiation-inducible ROCK kinase phosphorylates on FG-domain of NUP62 and increased oncogenic TF p63 nuclear traffic in squamous cancer cell published in EMBO Reports (doi: 10.15252/embr.201744523.), which is further introduced in News and View (EMBO Reports, 2018).

Response: We thank the reviewer for suggesting the articles which show that phosphorylation of Nup62, an FG-Nup, down regulates the active transport of TF p63 in squamous cancer cell. We have now cited these papers (see line 70 and 438 for citation numbers 30 and 31).

In addition, (ii) it could be worthy to discuss phosphorylation cascade from the view of cell biology. Since it sounds too artificial if all of potential sites get phosphorylation together. Activity of kinase or phosphatase depend on cellular condition.

Response: We agree with the reviewer. ERK phosphorylation cascade is responsible for FG-Nup phosphorylation targeting the S and T residues. The Phos_Max scenario considered in this study accounts for phosphorylation of all S, T, H and Y sites. Clearly, this scenario is not very relevant from a biology point of view as phosphorylation of all S, T, H and Y does not occur simultaneously in reality. The results of the Phos_Max scenario therefore serve as a theoretical limiting case of phosphorylated NPCs that feature a maximal phosphorylation-induced modification of charge and hydrophobicity. This has now been added in the last paragraph of the Introduction and in the second-to-last paragraph of the Discussion and Conclusions section.

Finally, (iii) Are phosphorylation sites considered in this study conserved in mammalian cells? It would be better to be shown or discussed.

Response: Clearly, some of the phosphorylation sites in yeast and mammals are conserved [37, 38] and many protein kinases in yeast have homologues in mammals. However, our computational simulations are aimed at exploring the fundamental mechanisms that drive changes in transport kinetics of phosphorylated NPCs, and are not meant to mimic specific biological conditions. We therefore choose not to discuss the biological implications of conservation in our paper.

Round  2

Reviewer 1 Report

The authors have nicely set and explained the limits to their study, so the reader are no longer led to possible misunderstandings. I have no further problems with the manuscript.